# Rheopheresis Performed in Hemodialysis Patients Targets Endothelium and Has an Acute Anti-Inflammatory Effect

**DOI:** 10.3390/jcm12010105

**Published:** 2022-12-23

**Authors:** Justine Solignac, Romaric Lacroix, Laurent Arnaud, Evelyne Abdili, Dammar Bouchouareb, Stéphane Burtey, Philippe Brunet, Françoise Dignat-George, Thomas Robert

**Affiliations:** 1Centre de Néphrologie et de Transplantation Rénale, Hôpital de la Conception, Assistance Publique-Hôpitaux de Marseille, 13005 Marseille, France; 2Center for CardioVascular and Nutrition Research (C2VN), Faculty of Medical and Paramedical Sciences, Aix-Marseille University, National Institute of Health and Medical Research (INSERM), National Research Institute for Agriculture, Food and Environment (INRAE), 13005 Marseille, France; 3Laboratoire de Biologie, Hôpital de la Timone, Assistance Publique-Hôpitaux de Marseille, 13005 Marseille, France

**Keywords:** apheresis, plasmapheresis, hemodialysis, microcirculation, endothelium dysfunction, cytokines, endothelial adhesion molecules, angiogenic factors, circulating endothelial cells

## Abstract

Background: Rheopheresis is a double-filtration plasmapheresis that removes a defined spectrum of high-molecular-weight proteins to lower plasma viscosity and improves microcirculation disorders. This technique can be performed in hemodialysis (HD) patients with severe microischemia. Interestingly, some studies showed that rheopheresis sessions improve endothelial function. Methods: Our study evaluated the inflammatory and endothelial biomarker evolution in 23 HD patients treated or not with rheopheresis. A *p* value ≤ 0.001 was considered statistically significant. Results: Thirteen HD patients treated by rheopheresis either for a severe peripheral arterial disease (N = 8) or calciphylaxis (N = 5) were analyzed. Ten control HD patients were also included in order to avoid any misinterpretation of the rheopheresis effects in regard to the HD circuit. In the HD group without rheopheresis, the circulating endothelial adhesion molecules, cytokines, angiogenic factor concentrations, and circulating levels were not modified. In the HD group with rheopheresis, the circulating endothelial adhesion molecules (sVCAM-1, sP-selectin, and sE-selectin) experienced a significant reduction, except sICAM-1. Among the pro-inflammatory cytokines, TNF-α was significantly reduced by 32.6% [(−42.2)–(−22.5)] (*p* < 0.0001), while the anti-inflammatory cytokine IL-10 increased by 674% (306–1299) (*p* < 0.0001). Among the angiogenic factors, only sEndoglin experienced a significant reduction. The CEC level trended to increase from 13 (3–33) cells/mL to 43 (8–140) cells/mL (*p* = 0.002). We did not observe any difference on the pre-session values of the molecules of interest between the first rheopheresis session and the last rheopheresis session. Conclusion: Rheopheresis immediately modified the inflammation balance and the endothelial injury biomarkers. Further studies are needed to understand the mechanisms underlying these biological observations.

## 1. Introduction

Rheopheresis is an apheresis treatment that removes—by 30 to 60% of their value—a defined spectrum of high-molecular-weight proteins (such as LDL-cholesterol, fibrinogen, a2-macroglobulin, immunoglobulin M [IgM], von Willebrand factor [vWF], and fibronectin) from human plasma, lowering blood plasma viscosity, as well as reducing erythrocyte and thrombocyte aggregation [1,2]. Rheopheresis is a double-filtration plasmapheresis, using membrane technology configured in a differential filtration array with two single-use membrane filters. By reducing crucial determinants of plasma viscosity [3], this technique aims to improve microcirculation disorders.

Microcirculation is the scene of interfaces among blood cells, plasma, and endothelial cells from the vascular wall. The quality of the exchanges among these components is essential for the proper oxygenation of tissues. Plasma viscosity and its determinants are major factors in capillary blood flow [3], but so in shear stress signaling and in endothelial cell activation status [4]. Modulating plasma viscosity therefore appears to be a relevant therapeutic target.

End-stage renal disease patients display both severe macrovascular and microvascular diseases, resulting in high cardiovascular morbi-mortality, not entirely explained by the traditional cardiovascular risk factors [5,6]. One way of understanding is the accumulation of uremic toxins, especially indolic uremic toxins, which are factors of microvascular aggression and impair the ability to repair injury [7,8,9,10,11]. These compounds are usually excreted by the kidneys, and are, for indolic uremic toxins, proteins that are bound so poorly and removed by dialysis. However, hemodialysis (HD) patients are characterized by a chronic inflammatory and pro-thrombotic state, reflected by increased levels of TNF-α, IL-6, IL-1β [12], sCD40L [13], vWF [14], fibrinogen [15], PAI-1 [16], tissue factor [17], and Factors VII and VIII [18]. Moreover, HD patients have increased levels of circulating adhesion molecules ICAM-1, VCAM-1, E-selectin, and P-selectin [19,20]. All these biological alterations can be related to endothelial dysfunction, described as a switch from protective endothelial phenotype toward a chronic pro-coagulant, pro-adhesive, and pro-inflammatory phenotype.

The rheopheresis effectiveness on microcirculation disorders has been demonstrated in a few clinical trials. According to the 2019 AFSA (American Society for Apheresis) guidelines, rheopheresis is a first-line treatment for dry age-related macular degeneration and sudden hearing loss [21]. Randomized controlled trials have indeed shown clinical efficiency of rheopheresis treatment in these indications [22,23]. Smaller clinical series evaluating rheopheresis in chronic limb-threatening ischemia resulting from peripheral arterial disease (PAD-CLTI) and calciphylaxis have shown promising results [24,25,26,27]. Our team is conducting a randomized controlled trial on PAD-CLTI in hemodialysis patients (RHEOPAD; Clinical trial number:03975946). Regarding tolerance, the addition of a second extracoporeal circulation increases the risk of hypotension, which is the major adverse event during a dialysis session. We reported the hypotension episode prevalence rate at 13.5% in a previous clinical series [28].

Interestingly, some studies showed that rheopheresis sessions led to an acute improvement of endothelial function. The control of vascular tone was indeed enhanced after rheopheresis treatment, assessed by flow-mediated vasodilation and skin vasodilator response to acetylcholine iontophoresis and to ischemia [29,30]. However, no studies have evaluated the biological effects of rheopheresis on endothelium dysfunction markers including inflammation mediators. Our study assesses the evolution of inflammation and endothelial lesion/regeneration markers in 23 HD patients treated or not with rheopheresis.

## 2. Material and Methods

### 2.1. Population

Thirteen HD patients treated by rheopheresis (HD group with rheopheresis) in the dialysis center of the Hospital of La Conception (Marseille, France) were consecutively included between 1 February 2018 and 1 September 2019. The indication for rheopheresis therapy was either PAD-CLTI or calciphylaxis. Ten HD patients not treated by rheopheresis (HD group without rheopheresis) with peripheral artery disease history and microinflammatory status, defined as CRP > 5 ng/mL without infection, were also included as control group. All patients received at least 3 weekly dialysis sessions. The data included in this study were anonymized, approved, and registered at the Health Data Portal of Assistance Publique-Hôpitaux de Marseille (PADS 22-370).

### 2.2. Rheopheresis Treatment

Rheopheresis technology was performed by the machine Plasauto Σ supplied by Hema.T Medical (Ramonville-Saint-Agne, France) and manufactured by Asahi Medical Co, Ltd. (Tokyo, Japan). The rheopheresis extracorporeal circuit has two filters one after the other, first a plasma separator (Plasmaflo 0P-05W[L]), which separates the plasma from the blood cells, and a second rheofilter (Rheofilter ER-4000), which separates high-molecular-weight plasma proteins from the plasma, with a theoretical threshold at 200 KiloDalton (KDa). Rheopheresis treatment was performed in tandem with the dialysis generator during the first one to two hours of the dialysis sessions (duration of rheopheresis depending on the plasma volume target) (Figure 1). No ultrafiltration was performed during tandem treatment. The patients started rheopheresis with a first phase with two weekly sessions (first and last dialysis session of the week) for 2 weeks, followed by a second phase with one weekly session (mid-week dialysis session) for at least 8 weeks. Then, treatment was continued once a week if the lesions were not healed. The plasma volume target was 40 mL/kg during the first phase and 60 mL/kg during the second phase.

### 2.3. Blood Assays

We performed blood assays immediately before (pre) and after (post) the following rheopheresis sessions: first, mid-protocol, and last sessions. Anticoagulation of the circuit was performed with citrate. As the control, we performed the same blood assays immediately before (pre) and after (post) the HD sessions without rheopheresis. In this case, anticoagulation of the circuit was performed with heparin.

Assays of circulating endothelial adhesion molecules (sVCAM-1, sICAM-1, sE-Selectin, and sP-Selectin), cytokines (TNF-α, sCD40-L, IL-1β, IL-6, IL-8, and IL-10), and angiogenic factors (Angiopoietin2, sEndoglin, and VEGF-A) were performed by the Mutliplex immunoassays: hAdhesion Mag LxPA 4-plex Kit from Bio-Techne, Milliplex Catalog ID.HCYTOMAG-C0K-09.Human Cytokine, and Catalog I0.1-IAGP1MAG-12K-04.Hu Angiogenesis/GF MAG 1 kits from Merck Millipore, respectively.

Circulating endothelial cells (CECs) were counted according to a previously published standardized protocol [31]. Briefly, the first 2 mL of drawn blood was discarded to avoid contamination by mural endothelial cells dislodged by venipuncture. CECs were isolated by immunomagnetic separation with magnetic beads (Dynabeads M-450, Thermo Fisher, Carlsbad, CA, USA) coated with CD146 (clone S-endo1, Biocytex, Marseille, France) and enumerated by fluorescence microscopy after acridine orange labeling at 490 nm. The stability of the microscope (Nikon Eclipse TE2000-S, Nikon, Tokyo, Japan) was regularly monitored with an Argo-Check Intensity LSG 130 (Argolight, Pessac, France). CECs were counted by an operator unaware of the patients’ clinical features; CECs were identified according to the following consensus criteria: rosette cell staining with acridine orange, size over 15 m, and bearing more than five beads.

Upper values of the reference interval provided by the laboratory based on heathy donors were at 300 ng/mL for sICAM-1; 2500 ng/mL for sVCAM-1; 50 ng/mL for sE-Selectin; 44 ng/mL for sP-Selectin; 140 pg/mL for sCD40L; 12 pg/mL for TNF-α; 3 pg/mL for IL-1β; 3.5 pg/mL for IL-6; 11 pg/mL for IL-8; 13 pg/mL for IL-10; 4,3 pg/mL for Angiopoietin 2; 7100 pg/mL for sEndoglin; 115 pg/mL for VEGF-A, and 20 cells/mL.

The measurements of the biological parameters after the apheresis and the dialysis session were corrected according to the modification of the plasma hemoconcentration. We used the Von Beaumont equation-based hematocrit measurement before (H1) and after (H2) the sessions to correct the values of the biological parameters by a factor (FC) to take into account the plasma volume reduction [32]: $\mathrm{FC}=\frac{H1\left(1-H2\right)}{H2\;\left(1-H1\right)}$.

We calculated the molecule concentration percentage change in patient blood measured immediately before (pre-value) and after (post-value) sessions as follows: $\frac{pre\;value-post\;value}{pre\;value}\;$.

### 2.4. Statistical Analysis

Continuous variables were expressed as medians [interquartile range]. Categorical variables were presented as a number (%). Values were rounded up to the tenth. We used a Student’s paired *t*-test when values were >30 and Wilcoxon paired test when values were <30 for comparative analyses. A *p* value ≤ 0.001 was considered statistically significant.

## 3. Results

### 3.1. Study Population

Baseline clinical and biological features of the HD group with rheopheresis and the HD group without rheopheresis are summarized in Table 1 and Table 2. Thirteen HD patients treated by rheopheresis were included, eight for chronic limb-threatening ischemia resulting from peripheral arterial disease (PAD-CLTI) and five for calciphyslaxis. HD patients treated by rheopheresis had diabetes (n = 8; 62%), hypertension (n = 13; 100%), coronary artery disease (n = 7; 54%), and peripheral artery disease (n = 9; 69%). They experienced a pronounced inflammatory syndrome with a median fibrinogen concentration of 5.9 g/L (5.2–6.5) and a CRP of 33.8 mg/L (14.0–47.5). The median CEC level at the inclusion was high at 16 cells/mL (2–34). Ten HD patients not treated by rheopheresis were included. Most had diabetes (n = 8; 80%), hypertension (n = 8; 80%), coronary disease (n = 6; 60%), and peripheral artery disease (n = 9; 90%). They displayed an inflammatory syndrome with a median CRP concentration of 7.9 mg/L (6.3–12.3). Compared to the HD group with rheopheresis, HD patients not treated by rheopheresis had a higher TNF-α concentration at the inclusion (*p* = 0.0008).

### 3.2. Acute Effect of Hemodialysis Sessions on Inflammatory and Endothelial Markers

In the HD group without rheopheresis, the post-HD, the circulating endothelial adhesion molecules, cytokines, angiogenic factor concentrations, and CEC levels were not modified (Table 3), except for sCD40L, which trended to increase significantly from 81.9 pg/mL (27.8–179) pre-HD to 670 pg/mL (195–889) post-HD (*p* = 0.002).

### 3.3. Acute Effect of Rheopheresis Sessions

In the HD group with rheopheresis, all patients experienced a reduction in the high-molecular-weight protein. The median percentage reduction in fibrinogen was at 53.4% [(−59.2)–(−45.4)], total cholesterol at 48.1% [(−55.4)–(−40.3)], triglycerides at 34.7% [(−43.4)–(−16.6)], LDLc at 68.3% [(−75.8)–(−59.8)], IgM at 54.1% [(−59.3)–(−46.1)], and α2-macroglobulin at 51.2% [(−62.9)–(−43.5)]. CRP was also reduced by 39.5% [(−48.1)–(−34.2)] (*p* = 0.01) (Table 4).

Circulating endothelial adhesion molecules (sVCAM-1, sP-selectin, and sE-selectin) experienced a significant reduction except sICAM-1 (Figure 2A and Table 4). Among the pro-inflammatory cytokines, TNF-α was significantly reduced by 32.6% [(−42.2)–(−22.5)] (*p* < 0.0001), while the anti-inflammatory cytokine IL-10 increased by 674% (306–1299) (*p* < 0.0001, Figure 2B and Table 4). Among the angiogenic factors, only sEndoglin experienced a significant reduction by −31.5% [(−43.1)–(−17.2)] (*p* < 0.0001, Figure 2C and Table 4). CEC levels trended to increase from 13 (3–33) cells/mL to 43 (8–140)cells/mL (*p* = 0.002, Figure 2D and Table 4).

### 3.4. Long-Term Effect of Rheopheresis Sessions

After a median of 18 (12–30) rheopheresis sessions, pre-rheopheresis fibrinogen concentrations were significantly reduced, from a median of 5.9 g/L (5.2–6.5) to 3.7 g/L (2.7–4.6) (*p* = 0.0007, Table 5). IL-10 trended to be reduced from 9.6 pg/mL (5.1–13.1) to 3.4 pg/mL (1.4–8.2) (*p* = 0.01, Table 5). We did not observe any difference between the values before the first session and before the last session concerning the other molecules of interest and the CECs.

## 4. Discussion

This is the first study to demonstrate that a rheopheresis session modifies the inflammation balance. Rheopheresis treatment lowered plasma viscosity, a major determinant of capillary blood flow, especially when vasomotor functions are impaired [1]. The rheopheresis effects on rheological parameters have been explored [1,2]; nevertheless, previous studies did not assess the effects on cytokines and endothelial biomarkers. We chose a stringent alpha risk at 0.001 to limit the statistical bias induced by the exploratory nature of our study [33,34]. We observed a clear immediate anti-inflammatory-induced rheopheresis process highlighted by the decrease in fibrinogen, CRP, and TNF-α and the increase in IL-10 at a median of more than six-fold the baseline value. The spectrum of inflammation and endothelial markers with statistically significant changes by rheopheresis sessions is summarized in Figure 3. We also observed that pre-rheopheresis fibrinogen is reduced in a long-term effect, which may reflect an improvement in the micro-inflammatory and pro-thrombotic state of our patients.

The rheopheresis session effects are not explained by the HD procedure. HD sessions had no effect on circulating endothelial adhesion molecules, cytokines, angiogenic factors, and CECs. This result is consistent with some studies in the literature [19,35,36,37]. The levels of sCD40L trended to increase after the dialysis sessions, but this needs to be confirmed on a larger sample size. Platelet activation by an extracorporeal circuit may explain the increase in sCD40L, which is a key factor in thrombostabilization and almost entirely produced by activated platelets [38,39].

The pulse of lowered inflammatory markers by the rheopheresis sessions is very quick, and the mechanism remains to be clarified. The rapid post-rheopheresis increase in IL-10, which can obviously not be removed by the rheofilter, indicates an immediate modulation of the immune system. The modification of inflammatory markers can occur through immediate mechanisms, such as shedding of soluble forms or release from vesicular stock, independently of gene expression. One hypothesis may be that the removal of LDL, and especially oxidized LDL, induces the anti-inflammatory signal. Interestingly, statin therapy exerts anti-inflammatory effects on innate and adaptive immune responses [40,41]. Similarly, some studies evaluating LDL apheresis performed in patients with familial hypercholesterolemia, have shown a decrease in TNF-α and soluble cell adhesion molecules and an increase in IL-10 [42,43,44,45].

It is well established that cytokines and circulating endothelial adhesion molecules represent a complex network interconnected with endothelial cell function [46]. These molecules are very interrelated, as suggested by the correlations in clinical studies [20,47,48]. Endothelial cells express cytokine receptors of TNF-α, IL-1, IL-6, and IL-8 and are able to self-produce cytokines [49]. Several studies showed that TNF-α and IL-1β induce the endothelial cell expression of VCAM-1, ICAM-1, E-selectin, and P-selectin, which shed in a soluble form in circulation. In turn, circulating endothelial adhesion molecules, thought to have chemo-attractant and angiogenic effects [50], but whose biological role has not been fully elucidated, stimulate various cytokine secretions in different cell types and perpetuate a chronic pro-inflammatory state [51,52].

Cytokines and circulating endothelial adhesion molecules reflect a state of endothelial cell activation, involving a switch from a quiescent phenotype to a pro-inflammatory one, and resulting in endothelial dysfunction [38,53]. These circulating molecules are considered as biomarkers for endothelial injury and vascular disease prognosis, including in hemodialysis patients [13,48,54,55]. Rheopheresis thus does target the endothelium and could have a beneficial effect on endothelial cells stress.

We observed that post-rheopheresis CEC levels trended to increase in our patients, reflecting an active endothelial phenomenon. Rheopheresis treatment could induce the reperfusion of the occluded capillary bed, releasing detached dead endothelial cells into circulation. It is well known that CEC levels are associated with cardiovascular disease severity and endothelial dysfunction [56,57] and are predictive of serious cardiovascular events, including in hemodialysis patients [58]. In our cohort with active vascular disease, pre-rheopheresis CEC levels remained stable, and the significance is unknown.

Our study presents some limitations. First, the outcomes are purely biological, and their consequences for patients are unknown. The data should be correlated with clinical endpoints. The effectiveness and tolerance outcomes were not collected. Second, it is a small cohort, and the results should be confirmed in a larger population, with assays performed in patients who are their own controls in order to obtain a strictly comparable population. The answer will be provided by the ongoing national project conducted by our team (RHEOPAD; Clinical trial number: 03975946). Third, interactions among the different mediators measured in the blood and the control of their production are complex. There is a myriad of mediators not measured in this study, or not yet discovered. Moreover, the values were measured at a point in time, at the end of the session, with no data on what occurs in the hours following the sessions. Apparently, the acute anti-inflammatory effect should therefore be taken with caution.

To conclude, regarding the biological effect, the immunologic effect is a new aspect of rheopheresis therapy, in addition to its purifying and rheological effects. Further studies are needed to understand the mechanisms underlying these biological observations.

## Figures and Tables

**Figure 1 jcm-12-00105-f001:**
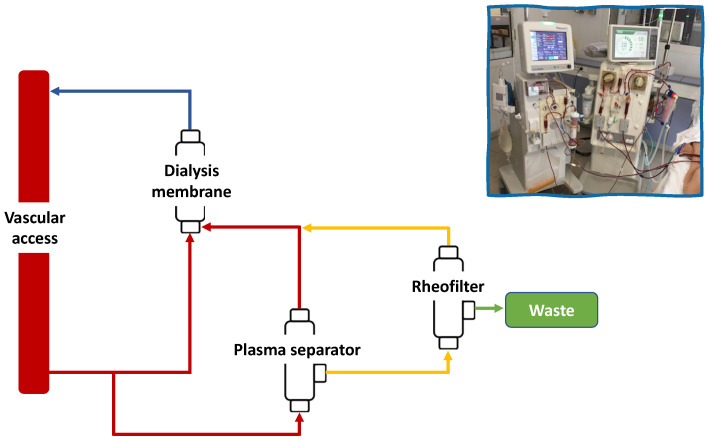
Diagram of the connection in tandem of the rheopheresis and dialysis extracorporeal circuit.

**Figure 2 jcm-12-00105-f002:**
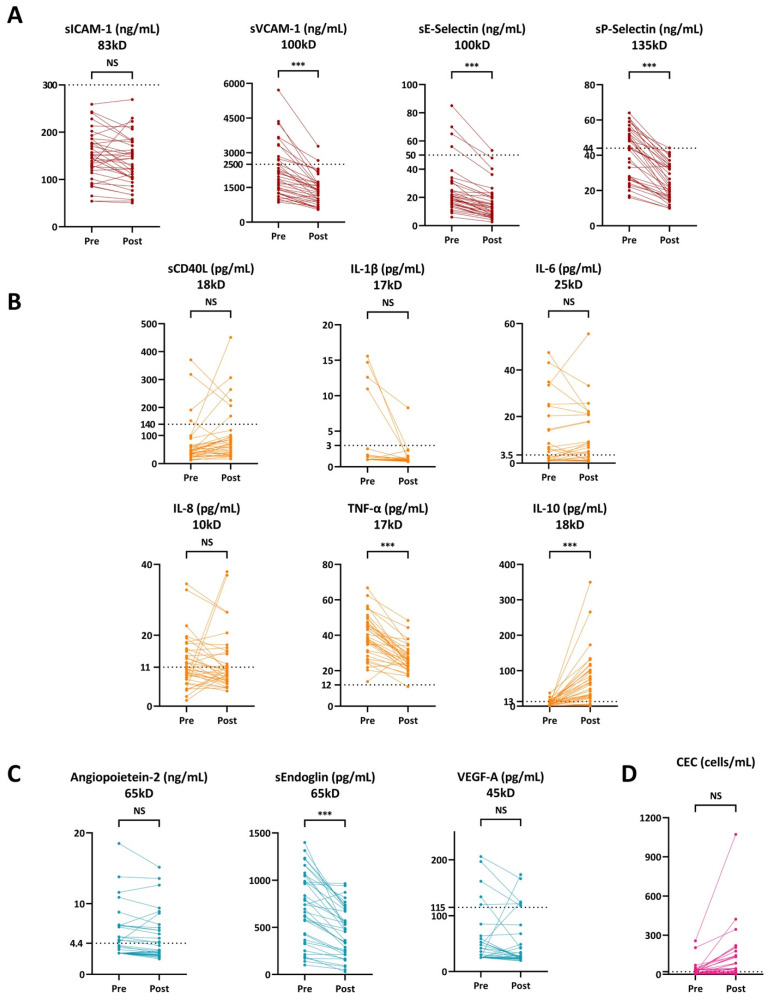
Circulating adhesion molecules (**A**), cytokines (**B**), angiogenic factors (**C**), and circulating endothelial cells (CECs) (**D**) before (pre-) and after (post-) rheopheresis sessions in the hemodialysis group with rheopheresis (N = 37). The dashed line represents the upper standard. NS: Non-statistically significant. ***: *p* < 0.001.

**Figure 3 jcm-12-00105-f003:**
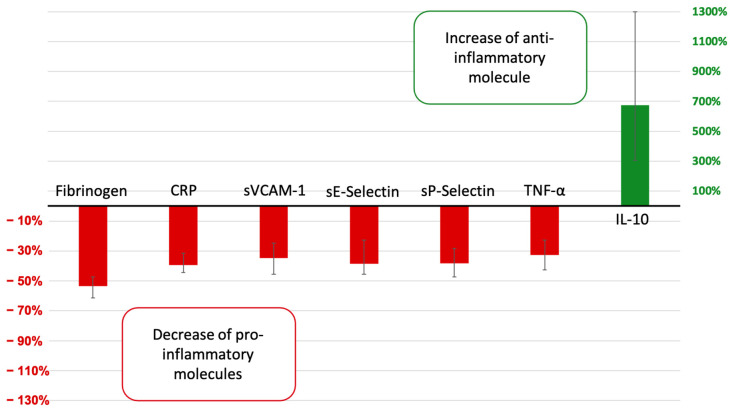
The spectrum of molecules with statistically significant changes after rheopheresis sessions. Changes are expressed by median percentage change with interquartile range. The two axes have different scales.

**Table 1 jcm-12-00105-t001:** Main clinical and biological characteristics.

	HD Group with Rheopheresis (N = 13)	HD Group without Rheopheresis (N = 10)
Age (years old)	68 (59–71)	78.5 (73.5–80.5)
Male gender	8 (62%)	8 (80%)
HD Vintage (Months)	33 (18–39)	20 (16–35)
BMI (kg/m^2^)	26.0 (23.2–29.5)	30.1 (27–71)
Diabetes	8 (62%)	8 (80%)
Hypertension	13 (100 %)	8 (80%)
Dyslipidemia	11 (85%)	6 (60%)
Smoking history	7 (54%)	7 (60%)
Coronary disease	7 (54%)	6 (60%)
Ejection fraction (%)	65 (53.7–65)	50 (50–62.5)
Peripheral occlusive arterial disease	9 (69%)	9 (90%)
Stroke history	0	1 (10%)
Arteriovenous fistula	9 (69%)	6 (60%)
Indication of rheopheresis:		-
PAD-CLTI	8	
Calciphylaxis	5	
Number of sessions	18 (12–30)	-
Albumin (g/L)	31.9 (28.4–35.2)	35.8 (33.2–39.5)

Quantitative variables are expressed as median [quartile 1–quartile 3]. HD: hemodialysis; BMI: body mass index; and PAD-CLTI: chronic limb-threatening ischemia resulting from peripheral arterial disease.

**Table 2 jcm-12-00105-t002:** Pre-session levels of endothelial adhesion molecules, cytokines, angiogenic factors, and CECs at the inclusion.

	HD Group with Rheopheresis (N = 13)	HD Group without Rheopheresis (N = 10)	*p* Value
Fibrinogen (g/L)	5.9 (5.2–6.5)	4.4 (4.3–5.6)	0.11
CRP (mg/L)	36.0 (12.0–49.0)	7.9 (6.3–12.3)	0.01
sICAM-1 (ng/mL)	243 (54.0–189)	220 (144–397)	0.1
sVCAM-1 (ng/mL)	1735 (1324–2150)	1146 (862–1442)	0.04
sE-Selectin (ng/mL)	25.0 (19.0–35.5)	34.0 (31.0–49.7)	0.36
sP-Selectin (ng/mL)	45.0 (27.5–53.0)	42.0 (40.0–46.7)	0.68
sCD40L (pg/mL)	46.3 (33.9–81.1)	82.0 (31.0–115)	0.68
IL-1β (pg/mL)	1.0 (1.0–1.5)	1.0 (1.0–4.6)	0.4
IL-6 (pg/mL)	6.2 (1.0–17.3)	1.0 (1.0–1.0)	0.01
IL-8 (pg/mL)	11.9 (8.7– 17.7)	9.0 (6.0–15.5)	0.23
TNF-α (pg/mL)	37.8 (29.0–45.1)	58.0 (50.0–62.0)	0.0008
IL-10 (pg/mL)	9.6 (5.1–13.1)	2.0 (1.0–7.5)	0.01
Angiopoietin 2 (ng/mL)	4.1 (3.0–6.9)	3.0 (3.0–3.9)	0.18
sEndoglin (pg/mL)	662 (574–971)	1347 (912–1455)	0.01
VEGF-A (pg/mL)	42.0 (25.0–56.0)	62.0 (25.0–104)	0.52
CECs (n/mL)	16 (2–34)	1 (1–4)	0.02

Quantitative variables are expressed as median [quartile 1–quartile 3]. Statistical test used was Mann–Whitney U. *p* < 0.001 was considered statistically significant. HD: hemodialysis; CRP: C-reactive protein; and CECs: circulating endothelial cells.

**Table 3 jcm-12-00105-t003:** Circulating adhesion molecules, cytokines, angiogenic factors, and CECs before and after dialysis sessions.

HD Group without Rheopheresis			
	Pre-Dialysis (N = 10)	Post-Dialysis (N = 10)	*p* Value
ICAM-1 (ng/mL)	222 (146–347)	193 (130–317)	0.04
VCAM-1 (ng/mL)	1146 (801–1556)	1120 (701–1629)	0.49
E-Selectin (ng/mL)	30.3 (26.6–45.6)	30.7 (24.5–45.4)	0.62
P-Selectin (ng/mL)	41.8 (39.8–48.7)	48.0 (46.3–64.2)	0.004
sCD40L (pg/mL)	81.9 (27.8–179)	669 (195–881)	0.002
IL-1β (pg/mL)	1.6 (1.0–4.6)	8.7 (0.9–11.6)	0.027
IL-6 (pg/mL)	1.0 (1.0–1.0)	0.9 (0.8–1.5)	0.57
IL-8 (pg/mL)	9.3 (5.0–17.4)	6.9 (4.2–16.6)	0.08
TNF-α (pg/mL)	57.8 (49.2–67.3)	38.8 (31.3–49.8)	0.009
IL-10 (pg/mL)	2.2 (1.0–7.5)	7.0 (1.7–9.8)	0.19
Angiopoietin (ng/mL)	3.0 (3.0–4.0)	2.68 (2.4–3.2)	0.04
sEndoglin (pg/mL)	1347 (873–1474)	1061 (766–1536)	0.19
VEGF-A (pg/mL)	59.8 (25.0–113)	114 (24.0–250)	0.43
CECs (n/mL)	1 (1.0–4.0)	1 (1.0–2.0)	0.13

Variables are expressed as median [Quartile 1–Quartile 3]. Values after dialysis sessions were corrected according to the hemoconcentration by Van Beaumont equation. Statistical test used was Wilcoxon test. *p* < 0.001 was considered statistically significant. HD: hemodialysis; and CECs: circulating endothelial cells.

**Table 4 jcm-12-00105-t004:** Inflammatory parameters, circulating adhesion molecules, cytokines, angiogenic factors, and CECs before and after rheopheresis sessions.

HD Group with Rheopheresis				
	Pre-Rheopheresis (N = 37)	Post-Rheopheresis (N = 37)	Percentage Change (%)	*p* Value
Fibrinogen (g/L)	4.72 (3.0–6.0)	1.9(1.4–2.7)	−53.4 [(−59.2)–(−45.4)]	<0.0001
CRP (mg/L)	15.3(5.2–36.9)	9.0 (2.8 –21.5)	−39.5 [(−48.1)–(−34.2)]	0.001
ICAM-1 (ng/mL)	148 (116–182)	132 (102–175)	−10 [(−20.7)–(−3.1)]	0.02
VCAM-1 (ng/mL)	1856 (1257–2707)	1330 (908–1654)	−34.7 [(−45.0)–(−23.2)]	<0.0001
E-Selectin (ng/mL)	21.0 (15.5–30.5)	12.3 (7.3–20.3)	−38.6 [(−54.8)–(−31.8)]	<0.0001
P-Selectin (ng/mL)	44.0 (26.5–53.0)	21.8 (16.1–33.8)	−38.3 [(−48.1)–(−29.1)]	<0.0001
sCD40L (pg/mL)	44.9 (31.3–64.3)	61.6 (34.3–96.1)	38.0 (7.7–104)	0.06
IL-1β (pg/mL)	1.0 (1.0–1.0)	0.9 (0.9–1.07)	−9.1 [(−22.1)–0]	0.04
IL-6 (pg/mL)	1.0 (1.0–11.3)	2.6 (0.9–13.5)	0 [(−12.4)–37.7]	0.93
IL-8 (pg/mL)	10.4 (7.7– 15.6)	9.6 (7.1–15.1)	−19.4 [(−32.1)–(−6.0)]	0.99
TNF-α (pg/mL)	39.1 (30.1–47.0)	25.7 (22.2–30.4)	−32.6 [(−42.2)–(−22.5)]	<0.0001
IL-10 (pg/mL)	5.6 (2.1–10.9)	60.9 (31.3–64.3)	674 (306–1299)	<0.0001
Angiopoietin2 (ng/mL)	3.5 (3.0–6.8)	3.3 (2.8–6.3)	−9.2 [(−14.8)–(−1.6)]	0.37
sEndoglin (pg/mL)	694 (353–1018)	474 (224–697)	−31.5 [(−43.1)–(−17.2)]	<0.0001
VEGF-A (pg/mL)	25.0 (25.0–56.0)	26.1 (23.7–46.0)	0 [(−29.0)–0]	0.6
CECs (n/mL)	13 (3–33)	43 (8–140)	317 (14.6–574)	0.002

Variables are expressed as median [Quartile 1–Quartile 3]. Values after rheopheresis sessions were corrected according to the hemoconcentration by Van Beaumont equation. Statistical test used was Student’s *t* test. *p* < 0,001 was considered statistically significant. HD: hemodialysis; CRP: C-reactive protein; and CECs: circulating endothelial cells.

**Table 5 jcm-12-00105-t005:** Inflammatory parameters, circulating adhesion molecules, cytokines, angiogenic factors, and CECs before the first and the last rheopheresis sessions.

HD Group with Rheopheresis			
	Pre-Rheopheresis First Session (N = 13)	Pre-Rheopheresis Last Session (N = 13)	*p* Value
Fibrinogen (g/L)	5.9 (5.2–6.5)	3.7 (2.7–4.6)	0.0007
CRP (mg/L)	36.0 (12.0–49.0)	11.5 (2.3–23.1)	0.12
ICAM-1 (ng/mL)	243 (54.0–189)	213 (54.0–159)	0.04
VCAM-1 (ng/mL)	1735 (1324–2150)	2485 (1172–3337)	0.05
E-Selectin (ng/mL)	25.0 (19.0–35.5)	18.0 (14.5–27.5)	0.009
P-Selectin (ng/mL)	45.0 (27.5–53.0)	44.0 (26.0–53.5)	0.69
sCD40L (pg/mL)	46.3 (33.9–81.1)	43.2 (29.9–75.2)	0.83
IL-1β (pg/mL)	1.0 (1.0–1.5)	1.0 (1.0–1.0)	0.99
IL-6 (pg/mL)	6.2 (1.0–17.3)	1.0 (1.0–5.4)	0.32
IL-8 (pg/mL)	11.9 (8.7– 17.7)	10.4 (6.2–14.4)	0.26
TNF-α (pg/mL)	37.8 (29.0–45.1)	40.4 (32.3–50.4)	0.73
IL-10 (pg/mL)	9.6 (5.1–13.1)	3.4 (1.4–8.2)	0.01
Angiopoietin2 (ng/mL)	4.1 (3.0–6.9)	3.0 (3.0–5.8)	0.71
sEndoglin (pg/mL)	662 (574–971)	740 (202.0–1117)	0.73
VEGF-A (pg/mL)	42.0 (25.0–56.0)	30.0 (25.0–53.5)	0.9
CECs (n/mL)	16 (2–34)	10 (2–27)	0.44

Variables are expressed as median [Quartile 1–Quartile 3]. Statistical test used was Wilcoxon test. *p* < 0.001 was considered statistically significant. HD: hemodialysis; CRP: C-reactive protein; and CECs: circulating endothelial cells.

## Data Availability

Data are not publicy archived.

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
