# Peer review of "Rheopheresis Performed in Hemodialysis Patients Targets Endothelium and Has an Acute Anti-Inflammatory Effect"

_jcm, 2022, doi:10.3390/jcm12010105_

Round 1
Reviewer 1 Report
Please add explanations of abbreviations, eg. PAD-CLTI
Describe more clearly the treatment schedule for rheopheresis in dialysis sessions.
Rosette cell staining with acridine orange, size over 15 m - explain the size of the cells.
The markings of the tables should clearly indicate whether they contain data of patients from the HD group treated with rheopheresis or from the untreated group.
Shouldn't statistical tests for related values be used?
Author Response
Please see the attachment.
C1: Please add explanations of abbreviations, eg. PAD-CLTI
R1: We thank the reviewer. We had indeed some missing abbreviations that we have defined as follows:
CEC in the method section “ Circulating endothelial cells”
PAD CLTI in the results section: “chronic limb threatening ischemia resulting from peripheral arterial disease (PAD-CLTI)”.
We have revised the Tables and figure legends as follows:
Table 1 “HD: hemodialysis; BMI: body mass index; PAD-CLTI: chronic limb threatening ischemia resulting from peripheral arterial disease.”
Table 2 and 5: “HD: hemodialysis; CRP: C-reactive protein; CEC : circulating endothelial cells.”
Table 3 and 4: “HD: hemodialysis; CEC: circulating endothelial cells.”
C2: Describe more clearly the treatment schedule for rheopheresis in dialysis sessions.
C2: We thank the reviewer and we added more precision in the method section as follows: “Rheopheresis treatment was performed in tandem with the dialysis generator during the first one to two hours of the dialysis sessions (duration of rheopheresis depending on the plasma volume target) (Figure 1). No ultrafiltration was performed during tandem treatment. The patients started rheopheresis with a first phase with two weekly sessions (first and last dialysis session of the week) for 2 weeks, followed by a second phase with one weekly session (midweek dialysis session) for at least 8 weeks. Then, treatment was continued once a week if the lesions were not healed. The plasma volume target was 40mL/kg during the first phase, and 60 ml/kg during the second phase.”
C3: Rosette cell staining with acridine orange, size over 15 m - explain the size of the cells.
R3: Sorry, but we don’t understand the meaning of this comment.
C4: The markings of the tables should clearly indicate whether they contain data of patients from the HD group treated with rheopheresis or from the untreated group.
R4: We thank the reviewer for this remark. We have specified the patient group at the first line of the table 3, table 4 and table 5 and in the Figure 2 legend.
C5: Shouldn't statistical tests for related values be used?
|
|
Correlation Coefficient |
||||||||||||||
|
ICAM-1 |
P-SEL |
VCAM-1 |
E-SEL |
Angiopoietin |
Endoglin |
VEGF-A |
IL-10 |
sCD40L |
Il-1 beta |
IL-6 |
IL-8 |
TNF alpha |
CEC |
||
|
P Value |
ICAM-1 |
|
0,121 |
0,259 |
0,144 |
-0,005 |
0,032 |
0,116 |
0,271 |
0,009 |
-0,144 |
0,06 |
0,537 |
0,029 |
-0,077 |
|
P-SEL |
0,477 |
-0,186 |
0,414 |
-0,381 |
-0,166 |
0,083 |
-0,254 |
0,266 |
0,212 |
-0,193 |
-0,02 |
-0,147 |
-0,248 |
||
|
VCAM-1 |
0,121 |
0,27 |
-0,135 |
0,039 |
0,367 |
0,121 |
0,532 |
-0,153 |
0,039 |
0,427 |
0,533 |
-0,059 |
-0,13 |
||
|
E-SEL |
0,397 |
0,011 |
0,426 |
-0,033 |
-0,079 |
0,115 |
0,134 |
-0,232 |
0,008 |
0,218 |
0,252 |
-0,096 |
-0,077 |
||
|
Angiopoietin |
0,976 |
0,02 |
0,817 |
0,847 |
0,246 |
0,121 |
0,228 |
-0,025 |
0,277 |
0,209 |
0,018 |
0,26 |
-0,053 |
||
|
Endoglin |
0,853 |
0,325 |
0,026 |
0,642 |
0,141 |
0,019 |
0,464 |
-0,15 |
0,285 |
0,364 |
0,246 |
0,248 |
0,135 |
||
|
VEGF-A |
0,493 |
0,627 |
0,474 |
0,497 |
0,477 |
0,91 |
0,317 |
0,227 |
0,163 |
0,389 |
0,157 |
0,345 |
0,021 |
||
|
IL-10 |
0,105 |
0,13 |
0,001 |
0,431 |
0,174 |
0,004 |
0,056 |
0,048 |
0,265 |
0,584 |
0,56 |
0,207 |
-0,149 |
||
|
sCD40L |
0,96 |
0,112 |
0,368 |
0,166 |
0,884 |
0,375 |
0,176 |
0,776 |
0,688 |
-0,219 |
0,042 |
0,463 |
-0,342 |
||
|
Il-1 beta |
0,397 |
0,207 |
0,819 |
0,964 |
0,096 |
0,087 |
0,336 |
0,113 |
2,55E-06 |
0,107 |
0,072 |
0,51 |
-0,462 |
||
|
IL-6 |
0,725 |
0,253 |
0,008 |
0,194 |
0,214 |
0,027 |
0,017 |
1,50E-04 |
0,193 |
0,527 |
0,386 |
0,07 |
-0,043 |
||
|
IL-8 |
0,001 |
0,908 |
0,001 |
0,132 |
0,917 |
0,142 |
0,354 |
3,14E-04 |
0,803 |
0,673 |
0,018 |
0,232 |
-0,091 |
||
|
TNF alpha |
0,865 |
0,386 |
0,728 |
0,572 |
0,12 |
0,138 |
0,037 |
0,219 |
0,004 |
0,001 |
0,681 |
0,1673 |
-0,093 |
||
|
CEC |
0,65 |
0,139 |
0,445 |
0,65 |
0,755 |
0,425 |
0,9 |
0,379 |
0,038 |
0,004 |
0,802 |
0,5937 |
0,585 |
||
R5: We thank the reviewer. We have been thinking about this question. In order to not multiply the number of statistical tests which are already very numerous, to keep a clear take home message throughout the article, we have chosen not to implement the correlation table below. We could put it as supplementary data if you consider it necessary.
Table S1 : Correlations of the pre-rheopheresis sessions values. Spearman's non-parametric correlation test. Correlations greater than 0.4 are in red color with corresponding p-values
Table S2 : Correlation of the post rheopheresis values. Spearman's non-parametric correlation test. Correlations greater than 0.4 are in red color with corresponding p-values.
|
|
Correlation coefficient |
||||||||||||||
|
ICAM-1 |
P-SEL |
VCAM-1 |
E-SEL |
Angiopoietin |
Endoglin |
VEGF-A |
IL-10 |
sCD40L |
Il-1 beta |
IL-6 |
IL-8 |
TNF alpha |
CEC |
||
|
P value |
ICAM-1 |
|
0,613 |
0,214 |
0,622 |
0,039 |
-0,358 |
-0,22 |
0,218 |
-0,184 |
0,226 |
0,052 |
0,355 |
0,051 |
-0,292 |
|
P-SEL |
0,0004 |
|
0,109 |
0,538 |
-0,025 |
-0,107 |
0,131 |
0,357 |
0,228 |
0,319 |
0,152 |
0,197 |
0,257 |
-0,287 |
|
|
VCAM-1 |
0,2648 |
0,5727 |
|
0,302 |
-0.125 |
-0,143 |
-0,118 |
0,438 |
-0,059 |
0,677 |
0,445 |
0,087 |
-0,165 |
-0,585 |
|
|
E-SEL |
0,0003 |
0,0026 |
0,111 |
|
0.144 |
-0,019 |
-0,093 |
0,491 |
-0,509 |
0,225 |
0,356 |
0,238 |
-0,13 |
-0,354 |
|
|
Angiopoietin |
0,8415 |
0,8959 |
0,517 |
0,4576 |
|
0,236 |
0,151 |
0,056 |
-0,189 |
0,196 |
0,062 |
0,382 |
0,189 |
0,036 |
|
|
Endoglin |
0,0671 |
0,5947 |
0,476 |
0,9266 |
0,173 |
|
0,143 |
0,142 |
-0,134 |
-0,05 |
0,249 |
0,162 |
0,148 |
0,122 |
|
|
VEGF-A |
0,2524 |
0,497 |
0,541 |
0,6326 |
0,371 |
0,413 |
|
0,443 |
0,141 |
0,018 |
0,393 |
0,02 |
0,233 |
0,149 |
|
|
IL-10 |
0,2563 |
0,0572 |
0,018 |
0,0068 |
0,742 |
0,416 |
0,006 |
|
0,044 |
0,317 |
0,588 |
0,311 |
0,078 |
-0,201 |
|
|
sCD40L |
0,3398 |
0,2343 |
0,76 |
0,0048 |
0,262 |
0,444 |
0,406 |
0,797 |
|
0,29 |
-0,102 |
0,055 |
0,339 |
-0,058 |
|
|
Il-1 beta |
0,2386 |
0,0921 |
5,45E-05 |
0,2407 |
0,246 |
0,775 |
0,917 |
0,056 |
0,081 |
|
0,228 |
0,299 |
0,096 |
-0,522 |
|
|
IL-6 |
0,788 |
0,4323 |
0,016 |
0,0583 |
0,715 |
0,15 |
0,016 |
1,31E-04 |
0,55 |
0,175 |
|
0,212 |
0,045 |
-0,185 |
|
|
IL-8 |
0,059 |
0,3068 |
0,655 |
0,2134 |
0,02 |
0,351 |
0,907 |
0,061 |
0,748 |
0,072 |
0,207 |
|
0,065 |
-0,011 |
|
|
TNF alpha |
0,7933 |
0,1778 |
0,393 |
0,5026 |
0,261 |
0,398 |
0,165 |
0,647 |
0,04 |
0,572 |
0,791 |
0,704 |
|
0,2 |
|
|
CEC |
0,1237 |
0,1318 |
0,001 |
0,0592 |
0,832 |
0,486 |
0,378 |
0,233 |
0,733 |
0,001 |
0,273 |
0,949 |
0,235 |
|
|

Reviewer 2 Report
In the present work Sulignac and collaborators evaluate the effects of rheoapheresis on molecule involved in endothelial function and responsible of the increased viscosity of the blood characteristic of microcirculation disorders. 13 HD patients treated with rheoapheresis were compared with 10 control HD patients. Their results show that rheoapheresis reduced significantly all circulating adhesion molecules but sICAM1. Among the inflammatory cytokines TNF was reduced while the antiinflammatory IL10 was incresed. Rheoapheresis also incresed the number of circulating endothelial cells. Apparently these acute changes did not last since the pre-session values ​​of these molecules were not different between the first and the last rheoapheresis session.
This study is quite interesting because it explores the mechanisms behind the improvement of the microcirculation of the procedure in hemodialysis patients. The limt of the study is the small sample and the lack of clinical outcome. It would be very interesting to know whether the rheoapheresis treatment improves at least the dialysis tolerance of the subjects that usually very poor. Furhermore it would be interesting to know whether the dialysis adeuquacy has been improved by the treatment.
the treatment with reoapheresis improved the dialysis tolerability of the subjects, which is often very reducedAuthor Response
We thank reviewer 2 for those comments. Our objective was, as pointed out by reviewer 2, to explore the changes of the indirect endothelial function biomarkers induced by the rheopheresis sessions. The clinical impact with robust clinical outcome will be provided by the national randomized controlled trial on PAD-CLTI in hemodialysis patients (RHEOPAD; Clinical trial number:03975946).
Regarding the dialysis tolerance, addition of a second extracoporeal circulation increase the risk of hypotension which the major adverse event during a dialysis session. We reported the hypotension episodes prevalence rate at 13.5% in a previous clinical series (DOI: 10.1002/jca.21955). Our manuscript objective is to describe the biological effect of the rheopheresis and we propose to not include this clinical information that we have already described.
Concerning the dialysis adequacy which is based on uremic toxin elimination during dialysis session, we did not compare the urea Kt/V between two group but will be an interesting area for a next manuscript.
We hope we have responded adequately to reviewer questions.
Other
After reviewing, we have changed the graphical abstract figure because some molecules were not translated from French to English.